# Genetic Adaptations of Soybean to Cold Stress Reveal Key Insights Through Transcriptomic Analysis

**DOI:** 10.3390/biology13110856

**Published:** 2024-10-23

**Authors:** Xiulin Liu, Chunlei Zhang, Sobhi F. Lamlom, Kezhen Zhao, Ahmed M. Abdelghany, Xueyang Wang, Fengyi Zhang, Rongqiang Yuan, Dezhi Han, Bire Zha, Wencheng Lu, Honglei Ren, Bixian Zhang

**Affiliations:** 1Soybean Research Institute, Heilongjiang Academy of Agriculture Sciences, Harbin 150086, China; liuxiulin1002@126.com (X.L.); zhangchunlei1989@yeah.net (C.Z.); zhaokz928@163.com (K.Z.); hljsnkywxy@163.com (X.W.); ddszhangfy2019@163.com (F.Z.); yrq18846121189@163.com (R.Y.); renhonglei2022@163.com (H.R.); 2Plant Production Department, Faculty of Agriculture Saba Basha, Alexandria University, Alexandria 21531, Egypt; sobhifaid@alexu.edu.eg; 3Crop Science Department, Faculty of Agriculture, Damanhour University, Damanhour 22516, Egypt; ahmed.abdelghany@agr.dmu.edu.eg; 4Heihe Branch Institute, Heilongjiang Academy of Agricultural Sciences, Heihe 164300, China; handezhi2008@163.com; 5School of Modern Agriculture and Ecological Environment, Resource Utilization and Plant Protection, Heilongjiang University, Harbin 150080, China; 15899078694@163.com

**Keywords:** soybean adaptation, transcriptome analysis, cold tolerance, gene expression, oxidative enzymes

## Abstract

This study examined the transcriptome and physiological responses of cold-tolerant soybean cultivar “KD52” and cold-sensitive cultivar “DS17”. Increased malondialdehyde levels demonstrated DS17’s vulnerability to lipid peroxidation and membrane breakdown. In contrast, cold exposure boosted antioxidant enzyme activity in the KD52 cultivar, including peroxidase and superoxide dismutase, demonstrating a strong antioxidant defense system against oxidative stress. A transcriptomic study mapped 54,532 genes and showed dynamic responses. This group had 234 differentially expressed genes (DEGs) that consistently changed over time, indicating unique expression patterns in the 2 cultivars. These key DEGs and physiological markers were linked to candidate genes for oxidative damage management and antioxidant defenses. The identification of improved defense mechanisms in KD52 and key genes should improve soybean cold stress tolerance.

## 1. Introduction

Soybean is an extremely valuable crop that serves many critical purposes for both humans and animals. It is the biggest global source of edible oil, meeting over 50% of demand, and is an important protein source [1]. While soybean yields have historically increased by 1.3% annually, it is uncertain whether this rate can be sustained considering the need to double current food production by 2050 [2]. Moving forward, efforts to increase soybean crop yields need to take into account and prepare for the stresses that will arise from climate change [3].

Cold stress is a major abiotic factor that negatively impacts plant growth and productivity at all developmental stages. It is expected to persist or intensify, prompting plants to evolve complex mechanisms to cope with abiotic stresses in order to survive, grow, and reproduce amidst yearly environmental fluctuations [4,5]. Cold stress disrupts metabolic processes, reduces photosynthetic efficiency, and increases oxidative stress through reactive oxygen species (ROS) accumulation. Understanding cold stress response mechanisms is essential for improving soybean and other crops’ resilience to this environmental challenge [6,7]. It also inhibits stomatal opening and causes a reduction in stomatal size, which further impairs photosynthesis efficiency [8]. Furthermore, it causes severe damage to cold-susceptible cultivars, leading to impaired chloroplast structures and reduced chlorophyll contents below the levels seen in cold-resistant cultivars such as sugarcane [9], wheat [10], barley [11], maize [12], peanut [13], cotton [14], and soybean [15,16]. Typically, plants do not exhibit cold resistance in the warm growing season, but as seasons change, they perceive the temperature drop in winter and subsequently develop freezing resistance [17]. The cold stress resistance of plants denotes their capacity to endure and sustain functionality after exposure to such low temperatures [18,19].

Although soybeans are typically classified as summer crops, cold stress can occur in early planting seasons or in regions with fluctuating temperatures. Specifically, in young soybean seedlings, cold stress significantly diminishes photosynthesis capability, with the impact being more pronounced under higher illumination [4]. Thus, cold tolerant varieties maintained higher photosynthesis efficiency by protecting the photosynthesis apparatus, which is a widely used indicator for cold-tolerant germplasm screening that needs to be determined in soybean. Cold acclimation in soybeans involves physiological and molecular adjustments that help mitigate the damage caused by sudden drops in temperature. Processes such as the up-regulation of cold-responsive genes, alterations in membrane fluidity, and accumulation of osmoprotectants are examples of how soybean cultivars may adapt to cold conditions. But these stresses can be lessened by a complex antioxidant defense system that controls the oxidative profile and boosts the activities of antioxidant enzymes like ascorbate peroxidase (APX), peroxidase (POD), catalase (CAT), and superoxide dismutase (SOD). When oxidative conditions are present, the accumulation of malondialdehyde (MDA) is an obvious indicator of membrane damage caused by free-radical chain reactions [20]. MDA is produced through the peroxidation of unsaturated fatty acids in the membrane, serving as a key indicator of oxidative stress-induced damage [21]. Plants generate ROS in response to cold stress and maintaining ROS homeostasis is crucial for regulating cold-responsive gene expression, with cold stress binding factors (CBFs) playing a key role in ROS detoxification [17,22,23,24]. Furthermore, it has been identified that the cold stress response pathway in rice includes a regulatory module mediated by ROS. To counteract the harmful effects of excessive ROS accumulation, plants have developed efficient antioxidant defense mechanisms involving enzymatic and non-enzymatic systems that scavenge ROS [17,22]. Researchers have identified several key gene families involved in ROS scavenging in soybeans, including SOD and glutathione peroxidase (GPX) [17]. The expression levels of the SOD and GPX genes are highly responsive to cold stress in soybeans. For instance, researchers found that cold stress conditions up-regulated the soybean copper/zinc superoxide dismutase (*GmCuZnSOD*) gene, and its overexpression could enhance oxidative stress tolerance in transgenic plants. Moreover, other ROS-related genes encoding NADPH oxidases, glycolate oxidases, and peroxiredoxins have also been reported to be differentially regulated in soybeans exposed to cold stress. The coordinated control of these ROS-scavenging genes is important for keeping the redox balance and letting soybean plants deal with cold stress [17,22].

In soybean (*Glycine max*), exposure to cold stress is a critical constraint that restricts the geographical range and limits productivity [6,7]. When subjected to low temperatures, soybean plants initiate a genetic response by regulating the expression levels of certain genes specifically involved in cold tolerance mechanisms [6]. Gaining insights into the molecular pathways that enable soybean to withstand cold conditions is essential for the development of soybean cultivars with improved cold tolerance traits. Recent research has uncovered several vital transcription factors (TFs) that play a central role in orchestrating soybean’s genetic response to cold stress [6,7]. Transcription factors that are highly conserved across plant species, known as CBFs, occupy a central role in the CBF-controlled cold response pathway [7]. For soybean, the majority of CBF genes (*GmDREB1s*) exhibit increased expression levels under cold stress, and overproduction of the *GmDREB1B;1* gene can improve cold tolerance when introduced into the model plant Arabidopsis. We have identified several other critically important transcription factors involved in soybean’s response to cold conditions, including *GmMYBJ1, GmNEK1, GmZF1, GmbZIP, GmTCF1a,* and *SCOF-1*. Transgenic Arabidopsis plants that express these transcription factors enhance cold tolerance by modulating the expression of genes specifically associated with cold response [6,7]. The CBF-dependent and CBF-independent pathways are interrelated and cooperate to control the expression of genes related to cold stress in soybeans. Some important post-transcriptional and post-translational systems, such as microRNAs and the ubiquitin–proteasome system, also help change how cold-responsive transcription factors work [22]. Elucidating these multi-layered regulatory networks can provide valuable insights for breeding cold-tolerant soybean varieties.

The transcriptome, comprising the entirety of transcripts within a cell during a specific developmental stage or physiological condition, has become a focal point for analysis with the emergence of next generation sequencing (NGS) and RNA sequencing (RNA-seq) technology [25,26,27]. RNA-seq, facilitated by comprehensive NGS platforms encompassing library construction, sequencing, and analysis systems [27], has revolutionized transcriptome analysis by precisely identifying differentially expressed genes (DEGs) and unraveling potential molecular mechanisms [25,26,27]. Over the past decade, researchers have widely applied RNA-seq to compare the transcriptomes of plants exhibiting cold tolerance and sensitivity under conditions of low-temperature stress [27,28,29]. Furthermore, transcriptomic analyses will continue to be a valuable methodology for elucidating the molecular underpinnings of cold stress responses across diverse plant species. Transcriptomics enables high-throughput screening of gene expression changes induced by low temperature exposure, which can enhance our understanding of cold tolerance mechanisms.

Although researchers have conducted transcriptomic and physiological studies on various crops to investigate cold stress tolerance, soybean has not yet extensively undergone such an integrative analysis. This study uses a new method by combining transcriptomic data with physiological markers to find the key genes that control oxidative stress responses and antioxidant defenses in different types of soybeans when they are exposed to cold stress. This integrative strategy, scarcely applied in soybean, aims to unravel the molecular basis of cold tolerance in this globally significant oilseed crop, addressing a significant knowledge gap. Here, RNA-seq analysis was employed to examine temporal changes in the transcriptome of cold-stressed leaf tissues from two soybean cultivars, KD52 (cold-tolerant) and DS17 (cold-sensitive). By investigating the dynamics of transcriptomic responses to cold stress in these cultivars, this study aims to uncover the regulatory mechanisms underlying soybean adaptation to cold stress.

## 2. Materials and Methods

### 2.1. Plant Materials

The Heilongjiang Academy of Agricultural Sciences conducted a study on cold stress soybean growth using the cold-tolerant genotype KD52 (Kedou52) and the cold-sensitive variety DS17 (Dongsheng17). Seeds were collected from Heilongjiang Academy of Agricultural Sciences, Harbin, Heilongjiang, China. The mature seeds of the two soybean cultivars (KD52 and DS17) were first sterilized by immersing them in 2% hydrogen peroxide (H_2_O_2_) for 30 min, followed by three thorough rinses with distilled water. The sterilized seeds were then placed in an incubator set at 20 °C with 18% relative humidity for 16 h to promote uniform germination. Once the plants reached the second compound leaf stage (V2), they were exposed to cold stress at 6 °C for 0, 2, 4, and 8 h to simulate cold injury. During the cold treatment, both the experimental and control groups were maintained under a controlled photoperiod of 16 h of light and 8 h of darkness. The temperature for the cold-stressed group was kept at 6 °C, while the control group was maintained at the optimal growth temperature of 20 °C. These conditions ensured a clear comparison between cold stress and non-stress environments

### 2.2. Determination of Physiological Indices

After cold stress treatments of 0, 2, 4, and 8 h, leaf samples were immediately flash-frozen in liquid nitrogen to halt metabolic activity and preserve biochemical integrity. Approximately 0.1 g of the frozen leaf tissue was ground to a fine powder using a chilled mortar and pestle. To assess the activities of key antioxidant enzymes, we measured superoxide dismutase (SOD) and peroxidase (POD) activities following established protocols. SOD activity was determined using the nitroblue tetrazolium (NBT) photoreduction method as described by [30]. The reaction mixture contained 50 Mm phosphate buffer (Ph 7.8), 13 Mm methionine, 75 Μm NBT, 2 Μm riboflavin, and 0.1 mL enzyme extract, with enzyme activity expressed in units per mg protein. POD activity was measured following [31], using guaiacol as a substrate. The reaction mixture consisted of 50 Mm phosphate buffer (Ph 6.0), 0.1% guaiacol, 0.03% hydrogen peroxide, and enzyme extract, with absorbance measured at 470 nm to determine the rate of guaiacol oxidation. Enzyme assays were performed using reagents sourced from Solarbio Company, Beijing, China.

Lipid peroxidation was quantified by determining malondialdehyde (MDA) content via the thiobarbituric acid (TBA) assay, based on the method of [32]. For this, approximately 500 mg of fresh leaf tissue was ground in liquid nitrogen and homogenized in 0.5 mL of 0.1% (*w*/*v*) trichloroacetic acid (TCA). The homogenate was centrifuged at 12,000× *g* for 15 min at 4 °C. Then, 1 mL of the supernatant was mixed with 2 mL of 20% TCA containing 0.6% TBA in a clean glass tube. The reaction mixture was heated at 95 °C for 30 min, followed by rapid cooling in an ice bath. After cooling, the samples were centrifuged again at 3000× *g* for 10 min at 4 °C, and the absorbance of the supernatant was recorded at 532 nm, 600 nm, and 450 nm using a UV-visible spectrophotometer. The MDA concentration was calculated using the formula described by [33], considering the extinction coefficient of 155 Mm^−1^ cm^−1^.

### 2.3. RNA Preparation and Sequencing

Leaf tissues from the two soybean cultivars, DS17 (cold-sensitive) and KD52 (cold-tolerant), were subjected to cold stress treatments at 6 °C for 0 (control, CK), 2, 4, and 8 h. After treatment, total RNA was extracted from the leaf tissues using Trizol reagent (Invitrogen, Carlsbad, CA, USA) following the manufacturer’s protocol. Each cold treatment, including the control, was replicated three times, resulting in a total of 24 RNA samples (2 cultivars × 4 time points × 3 biological replicates). RNA quality and integrity were first verified by running 1.0% agarose gel electrophoresis to check for intact ribosomal RNA bands. RNA concentration and purity were further assessed using a 2100 Bioanalyzer (Agilent Technologies, Santa Clara, CA, USA), with samples exhibiting an RNA integrity number (RIN) of greater than 7.0 being selected for subsequent analysis. Additionally, RNA purity was evaluated using a NanoDrop spectrophotometer (Thermo Fisher Scientific, Waltham, MA, USA), with A260/A280 ratios ranging between 1.8 and 2.0 considered acceptable.

For cDNA library preparation, total RNA (1 µg per sample) was used to construct libraries using the NEBNext UltraTM RNA Library Prep Kit for Illumina (NEB, Beijing, China). The process involved poly(A) mRNA enrichment, followed by fragmentation, first- and second-strand cDNA synthesis, adapter ligation, and PCR amplification. The quality of the constructed libraries was evaluated using the Bioanalyzer system to ensure that the average fragment size and library concentration met the required standards for sequencing. The sequencing of the prepared libraries was performed on an Illumina NovaSeq 6000 platform (Illumina Inc., Hercules, CA, USA), generating 150 bp paired-end reads. The raw sequence data were filtered to remove adapters, low-quality reads, and reads with ambiguous bases using Trimmomatic software (v0.39). The quality of clean reads was evaluated using FastQC (v0.11.9). Clean reads were aligned to the soybean reference genome (*Glycine max*, version Wm82.a2.v1) obtained from the NCBI database (https://www.ncbi.nlm.nih.gov/datasets/genome/GCF_004193775.1, accessed on 15 February 2024) using the STAR aligner (v2.7.10a) with default parameters. Uniquely mapped reads were retained for further analysis, and mapping efficiency was assessed through alignment statistics generated by STAR.

RNA-seq technology was employed to investigate transcriptome dynamics in the leaves of DS17 and KD52 under optimal temperature and cold stress. A total of 24 samples, consisting of 3 biological replicates at 4 time points (0, 2, 4, and 8 h after cold stress), were analyzed. The samples were designated for the cold-sensitive cultivar DS17 (DS17-0H, DS17-2H, DS17-4H, DS17-8H) and the cold-tolerant cultivar KD52 (KD52-0H, KD52-2H, KD52-4H, KD52-8H). A total of 24 RNA-sequencing libraries were generated from soybean leaf samples, with a total read count of 1.23 billion 150 bp paired-end reads per library and the number of sequenced reads per library ranged from 37.4 million to 83.2 million reads (Appendix A). After data filtration, about 1.17 billion clean reads were obtained, and the number of sequenced reads per library ranged from 35.7 million to 79.6 million reads, whereas the GC content was between 42.30 and 46.70% (Appendix A). The clean reads were submitted to the sequence read archive of the National Center for Biotechnology Information (NCBI) under BioProject ID No: PRJNA1128828. On average, about 94.02% of the high-quality reads (Q ≥ 30) were mapped to the soybean reference genome, ranging from 92.87% to 94.72%.

### 2.4. Differential Expression Analysis

To identify differentially expressed genes (DEGs) between the two soybean cultivars under cold stress, we performed differential expression analysis by first quantifying gene transcript abundances. Transcript expression levels were calculated using the fragments per kilobase of exon per million mapped reads (FPKM) method, which normalizes read counts based on transcript length and sequencing depth. Gene expression quantification was performed using RNA-Seq by expectation maximization (RSEM v1.3.2) (http://deweylab.biostat.wisc.edu/rsem/, accessed on 20 February 2024), which accurately estimates transcript abundances from RNA-Seq data. To account for biological variability and assess differential gene expression, we used the R package edgeR, which implements an empirical Bayes approach for the analysis of digital gene expression (DGE) data. This package enables the identification of DEGs by modeling count data, while correcting for sequencing depth and sample variability. We applied the false discovery rate (FDR) method to adjust *p*-values for multiple testing, as it helps control the expected proportion of false positives among the identified DEGs. Genes were considered significantly differentially expressed if they met the criteria of FDR < 0.2 and *p*-value < 0.05.

Additionally, redundancy analysis (RDA) and the Mantel test were used to explore relationships between the transcriptomic data and physiological traits measured under cold stress. These analyses were conducted using the vegan (v2.5-7) package in R (v4.0.5). Co-occurrence network analysis was performed to explore the interactions and relationships among DEGs under cold stress. We used Spearman’s correlation to generate correlation networks, where the line thickness corresponded to the strength of the correlation, line color represented the direction of regulation (up- or down-regulation), and dot size indicated the relative abundance of the gene in the network. These analyses provided insights into the key regulatory genes and interactions involved in cold stress adaptation.

### 2.5. Functional Enrichment Analysis

Gene annotation was performed using the gene ontology (GO) database (http://geneontology.org/, accessed on 20 February 2024) to classify the identified genes based on biological processes, molecular functions, and cellular components. Functional enrichment analysis of DEGs was carried out using the GO and KEGG pathway databases with clusterProfiler (v4.8.2), applying the Benjamini–Hochberg method (*p*.adjust = ‘BH’) to adjust *p*-values for multiple comparisons. Pathways and GO terms with adjusted *p*-value < 0.05 were considered significantly enriched, highlighting key biological processes, molecular functions, and pathways involved in soybean responses to cold stress.

### 2.6. qRT-PCR Analysis

Total RNA was extracted from leaf tissues of the two soybean cultivars, KD52 and DS17, using a previously reported method above to ensure high RNA integrity and purity. The extracted RNA was converted into complementary DNA (cDNA) using the PrimeScript™ RT reagent kit (Takara Bio Inc., Shiga, Japan) following the manufacturer’s instructions. For each reaction, approximately 1 µg of total RNA was reverse transcribed into cDNA. The relative expression levels of target genes were quantified through quantitative real-time PCR (qRT-PCR) using the SYBR Green I dye. The reaction mixture was prepared as follows: 10 µL of SYBR Green Premix Ex Taq (Takara), 1 µL of cDNA, 0.5 µM of forward primer, 0.5 µM of reverse primer, and nuclease-free water to a final volume of 20 µL. Each reaction was carried out in technical triplicates to ensure accuracy and reproducibility. The qRT-PCR amplification was conducted on a Bio-Rad CFX96 real-time PCR detection system under the following cycling conditions: 95 °C for 30 s for initial denaturation, followed by 40 cycles of 95 °C for 5 s and 60 °C for 30 s for amplification. To normalize the expression data and correct for variations in RNA input, the UBQ10 (ubiquitin 10) gene was used as the internal reference (housekeeping gene). The relative expression levels of the target genes were calculated using the 2^−ΔΔCt^ method, which compared the cycle threshold (Ct) values of the target genes to those of UBQ10. This allowed for the comparison of gene expression between the two cultivars and across the different cold treatment time points. The primers used for qRT-PCR analysis are listed in Appendix A. All primers were designed to span exon–exon junctions to avoid amplification of genomic DNA, and their specificity was confirmed by melt curve analysis and agarose gel electrophoresis of the amplified products.

### 2.7. Statistical Analysis

Data analysis was performed using SPSS version 26.0 (IBM, Armonk, NY, USA) to assess the physiological responses of the two soybean cultivars, KD52 and DS17, under different cold stress conditions. A one-way analysis of variance (ANOVA) was conducted to compare the means of three key physiological parameters—MDA, SOD, and POD—across four cold stress time points (0, 2, 4, and 8 h). Post hoc analysis was performed using Duncan’s multiple range test to evaluate significant differences between the mean values of the eight cold treatment groups (two cultivars across four cold treatments), where significance levels were set at *p* < 0.05. For each parameter, the results were expressed as the mean ± standard deviation (SD). Graphical representation of the results was performed using GraphPad Prism 9 to provide clear visualization of the significant differences between treatments, cultivars, and time points.

## 3. Results

### 3.1. Physiological Changes of Soybean Under Cold Stress

The physiological responses of the cold-sensitive soybean variety DS17 and the cold-tolerant variety KD52 to cold stress were evaluated by measuring malondialdehyde (MDA) levels and the activity of superoxide dismutase (SOD) and peroxidase (POD) at 0, 2, 4, and 8 h of cold exposure (Figure 1). The activity of MDA, a marker of oxidative damage, escalated substantially in both types throughout time. At 0 h, a notable disparity was evident between DS17 and KD52, with DS17 displaying elevated MDA levels, signifying that DS17 experienced greater oxidative damage prior to the application of cold shock. This tendency persisted over time, with MDA levels markedly rising in both kinds after 2 h, and DS17 consistently exhibiting more MDA activity than KD52 at 4 and 8 h. At the 8 h mark, MDA levels reached their zenith in both kinds, with DS17 exhibiting markedly greater oxidative damage.

The activity of the protecting enzymes POD and SOD were markedly elevated in the cold-tolerant cultivar KD52 for the entire cold stress period. At 0 h, POD activity was minimal in both kinds, with KD52 demonstrating somewhat greater activity than DS17. After 2 h, the augmentation of POD activity became markedly evident in KD52, leading to a substantial disparity between the two kinds. This trend persisted at 4 and 8 h, with KD52 exhibiting markedly greater POD activity than DS17. SOD activity exhibited a similar trend, with KD52 continuously demonstrating more activity than DS17 from the two-hour mark onward. At the 8 h mark, KD52 demonstrated the highest SOD activity, indicating a more robust enzymatic defense against oxidative stress relative to DS17.

### 3.2. Identification of Key Genes in Response to Cold Stress by Comparative Transcriptome Analysis

Differentially expressed genes (DEGs) between cold-treated and control (0 h) samples were identified for each cultivar between KD52 and DS17 at different time points (i.e., KD52-0H_vs._DS17-0H, KD52-2H_vs._DS17-2H, KD52-4H_vs._DS17-4H and KD52-8H_vs._DS17-8H) (Figure 2, Appendix A). A total of 54,532 genes were mapped in the genome, 2476 DEGs (581 up-regulated and 1895 down-regulated), 5271 DEGs (1837 up-regulated and 3434 down-regulated), 11,973 DEGs (2310 up-regulated and 9663 down-regulated), and 12,781 DEGs (5542 up-regulated and 7239 down-regulated) were identified in KD52-0H_vs._DS17-0H, KD52-2H_vs._DS17-2H, KD52-4H_vs._DS17-4H and KD52-8H_vs._DS17-8H (Figure 2a, Appendix A). To obtain the key differential genes, upset and flower diagrams were used to summarize the numbers of DEGs among the 4 different sets, and 234 DEGs were shared between the 2 varieties in the 4 different comparative pairs, including 112 up-regulated genes (Figure 2b) and 122 down-regulated genes (Figure 2c). A deeper exploration of the key DEGs in a different comparison group, which was detected primarily in samples at different times (0, 2, 4 and 8 h), revealed that the multiples of gene expression differences in the comparison groups changed under the cold stress (Figure 2d). Among the 234 DEGs, 42 key genes were identified based on the criteria of a log2FC > 1 and *p*-value < 0.05 for significant up-regulation, and a log2FC < −1 and *p*-value < 0.05 for significant down-regulation. These genes exhibited distinct patterns of expression, with the up-regulated and down-regulated genes prominently located at the top and bottom of the volcano plot, respectively (Figure 2d).

### 3.3. Functional Classification of DEGs and GO Analysis

The 234 key DEGs were assigned to 3 gene ontology (GO) categories: biological process, cellular component, and molecular function (Figure 3). In the biological process, important terms were associated with the sequestering of iron ions (GO:0097577), sequestering of metal ion (GO:0051238), phragmoplast microtubule organization (GO:0080175), cellular iron ion homeostasis (GO:0006879), maintenance of location in cell (GO:0051651), iron ion homeostasis (GO:0055072), male meiosis cytokinesis (GO:0007112), cytokinin biosynthetic process (GO:0009691), meiotic cytokinesis (GO:0033206), and maintenance of location (GO:0051235). In the cellular component, GO terms were associated with the extracellular matrix (GO:0031012). In molecular function, GO terms associated with ATP dimethylallyl transferase activity (GO:0052622), ADP dimethylallyl transferase activity (GO:0052623), tRNA dimethylallyl transferase activity (GO:0052381), and AMP dimethylallyl transferase activity (GO:0009824) were enriched (Figure 3a). Interestingly, our analysis revealed that the significantly enriched GO terms (FDR < 0.05) were associated with up-regulated gene enrichment. In total, 7 differentially expressed genes (DEGs) were significantly enriched in 15 gene ontology (GO) terms: FER2-1 (ferritin2C transcript variant X1), LOC100799520 (adenylate isopentenyl transferase 32C chloroplast), SFERH-4 (ferritin), LOC102663290 (kinesin-like protein KIN12A2C transcript variant X3), LOC100802179 (vacuolar iron transporter homolog 3), LOC100817437 (kinesin-like protein IN12A2C transcript variant X1), and LOC100777757 (unknown) (Figure 3b). Among the significant GO terms, “sequestering of iron ion” (GO:0097577) in biological processes and “ATP dimethylallyl transferase activity” (GO:0052622) in molecular functions were the most prominent. The expression levels of the 42 DEGs were higher in KD52 compared with DS17, suggesting their crucial role in cold stress tolerance in soybean.

### 3.4. Association Analysis Between Key DEGs and Physiological Indices

We conducted association analysis (Figure 4) to uncover relationships between the two cultivars under the four cold stress treatments with different physiological parameters (POD, SOD, and MDA). The findings showed that redundancy analysis (RDA) clearly showed the difference between the cold-tolerant cultivar KD52 and the cold-sensitive cultivar DS17 (Figure 4a). The RDA biplot shows connections by lining up the cold-tolerant KD52 cultivar with the POD and SOD vector directions, which suggests that there are positive correlations. Conversely, the DS17 cultivar, which is susceptible to cold, was correlated with the MDA vector, indicating a strong connection with membrane degradation. The cultivar KD52 showed a positive correlation with POD (r^2^ = 0.128, *p* = 0.247) and SOD (r^2^ = 0.350, *p* = 0.008), indicating a remarkable association between antioxidant gene activation and enzymatic activity. In contrast, the DS17 cultivar significantly and positively correlated with MDA levels (r^2^ = 0.540, *p* = 0.001), linking membrane damage with altered gene expression.

To further relate transcriptional and physiological responses to cold stress, correlation analysis was performed between those 42 key DEGs identified earlier and activity levels of the antioxidant enzymes SOD and POD along with the membrane damage marker MDA (Figure 4b). The heatmap visualization displays Pearson correlation coefficients between each DEG and physiological indicator, as distinct clusters of correlated genes are evident. Several DEGs showed strong positive correlations with SOD and POD, linking them to the up-regulation of antioxidant defenses during cold stress. Meanwhile, a subset of DEGs was highly negatively correlated with MDA, suggesting possible roles for these genes in maintaining membrane stability under cold conditions. Among all DEGs, 19 up-regulated DEGs significantly negatively correlated with MDA, 2 DEGs (LOC100776078 and LOC100817437) positively correlated with POD, and 5 DEGs (LOC106795140, LOC102664221, LOC100776078, LOC100817437, and LOC100798870) were positively correlated with SOD. The 21 down-regulated DEGs significantly positively correlated with MDA, and 6 DEGs (LOC100816293, LOC100818493, LOC100786381, LOC102666666, LOC102662268, and LOC100819021) significantly negatively correlated with POD and SOD (Figure 4b,c).

For a deeper investigation of the associations between up- and down-regulated DEGs and the physiological parameters MDA, POD, and SOD during the cold stress response in soybean, a co-expression network analysis was performed (Figure 4c). Interestingly, we primarily linked MDA to up-regulated DEGs. This suggests that the increased expression of these genes correlates with higher MDA levels and greater membrane damage from cold stress. Conversely, the complex transcriptional regulation involving both activation and repression of genes interconnects the antioxidant enzymes POD and SOD with both up-regulated and down-regulated DEGs. The Sankey diagram depicting the correlations between physiological indices, their association with DEGs, DEG functions, and DEG regulation (up and down) under cold stress across four cold treatments is shown in Figure 4d. It gives an overview of the correlations between the three physiological indicators and DEGs under cold stress throughout four cold treatments. Moreover, it illustrates the strong correlation between physiological changes and changes in gene expression for various biological processes during the cold stress exposure period. The data revealed that DEGs were most active at 2 and 4 h. These DEGs’ expressions significantly increased over the 8 h cold stress period. We identified several hormone-responsive transcription factors as key DEGs.

### 3.5. Gene Expression Profiles over Graded Cold Stress Treatment Durations

To further investigate the transcriptional response of two soybean cultivars (KD52 and DS17) to cold stress, we performed RNA expression analysis on samples subjected to 0, 2, 4, and 8 h of cold treatment. Figure 5 and Appendix A show the relative expression levels of 17 genes across 2 soybean cultivars (KD52 and DS17) subjected to 0 h (CK), 2 h, 4 h, and 8 h of cold stress. In general, the data demonstrated fine-tuned transcriptional adjustments over prolonged exposure to low temperatures in a cultivar-dependent manner. Also, the expression profiles did not always clearly correlate with a single factor, as both induced and variable/non-induced expression patterns were observed under cold stress. For example, a subset of genes displayed maximal relative expression at 2 h, specifically under the KD52 cultivar cold treatment. These included LOC100799520, SFERH-4, LOC102663290, LOC106795140, LOC100500242, LOC100796525, LOC100817437, and LOC102668758. For these genes, the cold-induced transcriptional up-regulation peaked rapidly at 2 h, but only in KD52, as their expression remained lower in DS17 across all time points. This early response for KD52 suggests that these genes may be involved in cold stress mechanisms that work faster or more strongly in KD52 than in DS17. In contrast, the two genes, LOC100500496 and LOC100819021, had similar responses, but with the other cultivar, DS17. The expression data identified a subset of genes with peak relative expression in the control (CK) samples: LOC100777757, LOC100791780, LOC113001461, LOC102668758, LOC100801846, and LOC547706. These genes showed the highest baseline expression levels in the unstressed control condition for both DS17 and KD52 cultivars. Their expression then decreased with cold treatment over time. This implies that these genes participate in down-regulated processes or pathways during the cold response. Their higher basal expression in the absence of cold stress suggests that housekeeping roles are unrelated to cold adaptations.

Consequently, we conducted clustering heatmap visualization using the relative expression levels across all genes for the two cultivars subjected to 0, 2, 4, and 8 h of cold treatment (Figure 6). The results indicated distinct patterns of up- and down-regulation associated with cultivar, treatment duration, and their interaction. The heatmap clustering consisted of three distinct clusters: C1, C2, and C3. In cluster C1, there were seven genes—LOC100799520, LOC100791780, LOC100796525, LOC100500242, LOC106795140, LOC100817058, LOC100817437, and LOC113001461—that were mostly turned on in KD52 after 2–8 h of cold stress compared with DS17. This strong activation suggests that these genes are very important in KD52’s early cold response mechanisms. They may help with things like changing the membrane, activating transcription factors, and breaking down carbohydrates, all of which protect against low temperatures.

The expression levels of eight genes in Cluster C2—SFERH-4, LOC102663290, LOC100777757, LOC102668758, LOC100803418, LOC100500496, LOC100801846, and LOC547706—increased slowly over time in both cultivars from 0 to 8 h, though they stayed higher in KD52. These genes may be involved in general adaptation pathways activated as cold exposure is prolonged, including circadian rhythm adjustment, osmolyte synthesis, and reconfiguration of metabolism. Notably, Cluster C3 comprised only one gene, LOC100819021, which displayed a distinct expression profile compared with other highlighted genes. While many genes showed maximal expression either in KD52 or both KD52 and DS17, LOC100819021 exhibited a specific up-regulation at the control and 4 h timepoints. In addition to gene clusters, we divided the treatments into two major cultivar groups, each exhibiting a distinct divergence in cold responses based on genetic background. This highlights the significant role of genotype effects in directing gene activity and reprogramming under stress.

## 4. Discussion

Low temperatures are a significant environmental condition that not only restricts plant growth, development, and productivity but also endangers food and nutrition security [34]. Our research provides insight into the adaptive processes that make soybean tolerant to cold stress by exposing the genes and pathways activated during this crop’s response to cold stress. The cold’s effects should not be underestimated due to climate uncertainty in production [35,36]. Cold stress disrupts the balance between ROS generation and metabolism in plants [37]. When exposed to low temperatures, plants may produce reactive oxygen species (ROS)—such as superoxide, hydrogen peroxide, and hydroxyl radicals—at rates exceeding their detoxification capacity. This ROS accumulation can lead to cellular damage and, if severe, plant death in response to cold and other abiotic stresses [38].

Our research examined the concentrations of MDA, SOD, and POD enzymes in the leaves of two soybean cultivars, DS17 and KD52, under cold environmental conditions. MDA is a result of ROS oxidation of lipids, which indicates oxidative damage. On the other hand, SOD and POD are enzymes with antioxidant properties that assist in reducing the impact of ROS. Moreover, during cold stress, the cultivar DS17 accumulated substantially more MDA than the cultivar KD52, suggesting that DS17 is more susceptible to oxidative damage caused by ROS, leading to the peroxidation of membrane lipids [39,40,41]. However, KD52 had higher levels of SOD and POD, indicating a stronger antioxidant system to get rid of ROS and protect cells. The elevated MDA levels seen in DS17 indicate vulnerability, while KD52 stimulates the release of defensive antioxidants. Such distinct response patterns observed between the two cultivars indicate the utilizing of different physiological mechanisms to mitigate cold-induced oxidative stress [39,42].

Crops in various regions worldwide face extreme low temperatures [43,44,45,46,47], and also encounter limited water availability due to drought or disrupted water movement and uptake in cold conditions [48]. There is limited knowledge regarding how plants are affected by both chilling and drought stresses, and it is uncertain if their responses are distinct or similar. Plants can show similar molecular and physiological reactions when exposed to cold and dry conditions (common response), while some responses may be unique to a particular stressor [49,50]. Exposing plants to both low temperature and drought at the same time can hinder plant growth, which ultimately and negatively impacts productivity [51]. Research indicates that the effects of chilling and drought stress on stomatal development and leaf growth are similar, but the physiological processes affected by drought differ from those affected by chilling [52]. Typically, chilling stress reduces the speed of various physiological and metabolic processes in plants from a thermodynamic perspective [53]. It also significantly decreases germination speed and consistency, impairs seedling strength, and slows down plant growth progress [54], leading to substantial reductions in crop production [43].

RNA sequencing was performed on 24 RNA libraries in this study to identify DEGs between the cold-tolerant cultivar KD52 and cold-sensitive cultivar DS17 at 0, 2, 4, and 8 h of cold exposure. The comprehensive analysis of DEGs identified 234 key genes shared by the 4 comparative pairs, indicating a dynamic response to such cold stress [55]. Out of the 234 key DEGs identified in response to cold stress, 112 showed up-regulation and 122 demonstrated down-regulation. These changes in gene expression profiles highlight the complex nature of the soybean’s response to cold stress, impacting various distinct pathways. [56]. Previous research has similarly underscored the intricate regulatory networks that govern plant reactions to abiotic stressors [28,29].

Furthermore, the soybean cold stress response is linked to a variety of biological processes, cellular elements, and molecular activities [56,57,58]. The gene ontology (GO) analysis sheds light on these molecular activities and important pathways. Several important terms were found in the biological process category: phragmoplast microtubule organization (GO:0080175), iron ion sequestration (GO:0097577), and cytokinin biosynthesis process (GO:0009691). Previous studies have reported the importance of these processes in mediating responses to different environmental stresses [59,60,61,62]. The extracellular matrix was linked to the cellular component category (GO:0031012), highlighting the significance of cellular structure and organization in the body’s reaction to cold stress. The molecular function category contained terms that suggested the involvement of these genes in nucleotide binding and transferase activities, such as ATP dimethylallyl transferase activity (GO:0052622) and ADP dimethylallyl transferase activity (GO:0052623). Abiotic stress responses in plants have been the subject of studies that have produced similar findings [63,64]. These GO terms’ significance in soybean response to cold stress is highlighted by the up-regulation of DEGs linked to them. A significant enrichment in 15 GO terms was observed for 7 DEGs—FER2-1, LOC100799520, SFERH-4, LOC102663290, LOC100802179, LOC100817437, and LOC100777757—comprising a concentrated set of key genes associated with cold stress adaptation. Among these, the plant response to oxidative stress and iron homeostasis has been linked to ferritin (FER2-1). The significant enrichment of the term “sequestering of iron ion” (GO:0097577) suggests that the regulation of iron ion homeostasis is important for soybean response to cold stress. Furthermore, the significance of nucleotide binding and transferase activities in this response is highlighted by the involvement of ATP dimethylallyl transferase activity (GO:0052622). Finally, by highlighting the importance of particular biological processes, cellular elements, and molecular functions, this study adds significant knowledge about the molecular mechanisms of soybean cold stress response. The cold-sensitive cultivar KD52 exhibits up-regulation of key DEGs linked to these GO terms, indicating their possible function as molecular markers or targets for improving soybean resistance to cold stress.

Using RDA and Mantel tests in our research has helped us learn a lot about the complicated relationships between physiological markers (MDA, POD, and SOD) and DEGs in DS17 and KD52 soybean cultivars under cold stress conditions. Our findings show clear correlations and provide insight into how each cultivar reacts differently to harsh environmental circumstances. Our research found that MDA was a significant factor that correlated with DS17, indicating that MDA plays a crucial role in this cultivar’s reaction to cold stress. On the other hand, KD52 showed strong connections with both POD and SOD, showing that both POD and SOD play a role in reducing the effects of cold stress. The Mantel tests further deepened our understanding of the relationship between key DEGs and physiological indices. The observed negative correlation between 19 up-regulated DEGs and MDA suggests their potential role in mitigating lipid peroxidation, a common consequence of environmental stress. Also, the fact that two DEGs (LOC100776078 and LOC100817437) were positively correlated with POD and five DEGs (LOC106795140, LOC102664221, LOC100776078, LOC100817437, and LOC100798870) were positively correlated with SOD suggests that they are involved in antioxidative processes. This supports the antioxidant defense mechanisms that are activated in response to cold stress. Conversely, the positive relationship between 21 DEGs that were down-regulated and MDA suggests that lipid peroxidation-reducing processes may have been down-regulated, which is what led to the positive relationship with MDA. The negative correlation of six DEGs (LOC100816293, LOC100818493, LOC100786381, LOC102666666, LOC102662268, and LOC100819021) with both POD and SOD further highlights their potential roles in modulating ROS levels, emphasizing their significance in the cold stress response.

The findings of this study demonstrate significant variations in gene expression patterns between the cold-tolerant cultivar KD52 and the cold-sensitive cultivar DS17 during differing periods of cold stress treatment. The observed disparities indicate that the early and prolonged activation of cold-responsive genes by KD52 is crucial for its ability to tolerate cold temperatures. By contrast, DS17 displays delayed and less robust transcriptional responses, which could explain its heightened vulnerability to cold stress. A significant finding is the swift increase in expression of many genes related to the protection against oxidative stress, maintenance of iron ion balance, and organization of the cytoskeleton in KD52 cells during the initial 2 h of exposure to cold temperatures. The prompt and strong activation of genes such as LOC100817437 and LOC102668758 in KD52 is consistent with prior research indicating that early transcriptional activation is essential for the development of cold stress tolerance in different cereal crops. For example, Nakashima et al. [65] highlighted the significance of early activation of transcription factors in rice and Arabidopsis when exposed to abiotic stress. This activation is crucial for initiating subsequent protective processes, such as the synthesis of antioxidant enzymes. Furthermore, other investigations showed that the timely activation of genes that respond to cold conditions, such as CBF/DREB1 transcription factors, is crucial for the process of cold adaptation in wheat [66,67].

Conversely, the delayed gene expression responses of DS17 may indicate inadequate or slower signaling, leading to reduced efficacy in protecting against oxidative stress and membrane damage. The down-regulation of specific genes in DS17, which are expressed at higher levels in KD52 during cold stress, provides further evidence of variations in transcriptional repression mechanisms between the two types. Prior investigations have confirmed previous findings, showing that plant cultivars that are susceptible to cold temperatures frequently show reduced suppression of genes that respond to stress after being exposed to low temperatures. Their heightened susceptibility to cold and overall lower cold tolerance may be attributed to metabolic imbalances resulting from this poor gene regulation [68,69].

## 5. Conclusions

This work elucidates the molecular basis of cold stress responses in soybean. Based on the physiological data, the cold-tolerant KD52 cultivar seems to have better antioxidant defenses, with higher levels of SOD and POD activity to lower oxidative damage. MDA levels are higher in cold-sensitive DS17, indicating membrane damage. A comparative transcriptome study identifies 234 DEGs that change across cold treatment timepoints, with differing temporal expression patterns in the two cultivars. An analysis of GO shows how iron homeostasis, microtubule architecture, the extracellular matrix, and nucleotide binding are involved in the response of soybeans to cold stress. DEG expression and physiological indicators suggest that antioxidant-modulating genes regulate oxidative and membrane damage. Genes such as LOC100799520 and SFERH-4 exhibit rapid induction in KD52, indicating their rapid activation of protective mechanisms. Commonly elevated genes, with a stronger increase in KD52, may facilitate adaptation responses during prolonged cold. This research explains cold acclimation networks and finds soybean genes that may be cold-resilient. Functional analysis of these transcriptional regulators may reveal cold tolerance genes. Molecular breeding or biotechnology may use the identified genes and pathways to enhance soybean cold adaptability. However, this study provides significant insights into the molecular mechanisms underlying the response of soybean plants to cold stress, and it is crucial to conduct further research to gain a deeper understanding of the roles of transcriptional regulators and their target genes in cold tolerance. Moreover, exploring the potential of the identified cold-responsive genes and pathways for molecular breeding or biotechnological approaches could pave the way for developing soybean cultivars with enhanced cold tolerance. Such efforts would contribute to improving soybean productivity under increasingly unpredictable climatic conditions, thereby enhancing food security and agricultural sustainability.

## Figures and Tables

**Figure 1 biology-13-00856-f001:**
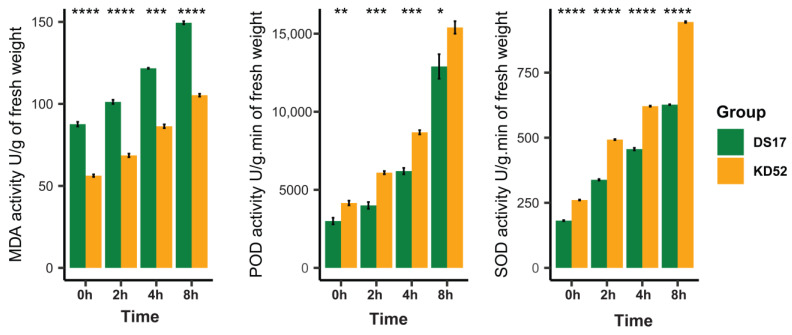
Physiological responses of cold-sensitive soybean variety DS17 and cold-tolerant variety KD52 under cold stress. Levels of damage indicators malondialdehyde (MDA) and protective enzymes superoxide dismutase (SOD) and peroxidase (POD) were measured at 0 (CK), 2, 4, and 8 h after cold treatment. Statistical analysis showed differences in physiological responses between DS17 and KD52 varieties at different time points during cold stress. *, **, ***, and **** indicate significant difference at *p*-value of <0.05, <0.01, <0.001, and <0.0001, respectively.

**Figure 2 biology-13-00856-f002:**
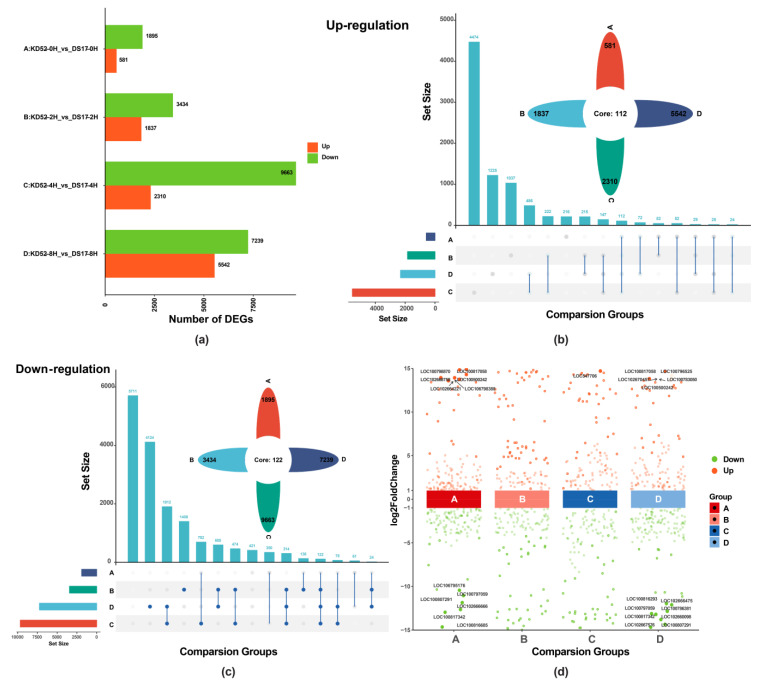
Identification of key genes in response to cold stress by comparative transcriptome analysis. (**a**) DEGs between DS17 and KD52 varieties of soybean at different times (0, 2, 4, and 8 h) during cold stress. (**b**,**c**) Upset and flower graphs showing overlapped DEGs (up- and down-regulated genes) among different time points (0, 2, 4, and 8 h after the initiation of cold treatment). Letters from A, B, C, and D indicate the four comparison groups KD52-0H_vs._DS17-0H, KD52-2H_vs._DS17-2H, KD52-4H_vs._DS17-4H and KD52-8H_vs._DS17-8H, respectively. A total of 234 DEGs (112 up-regulated (**b**) and 122 down-regulated (**c**)) in common. (**d**) Key differential gene expression analysis shows significantly up- and down-regulated genes across all four comparison groups. A log2FC > 1 and *p*-value < 0.05 is indicated as significantly up-regulated in red, while alog2FC < −1 and *p*-value < 0.05 is indicated as significantly down-regulated in green. A total of 42 key genes, which are located at the top and bottom of the volcano plot, were obtained.

**Figure 3 biology-13-00856-f003:**
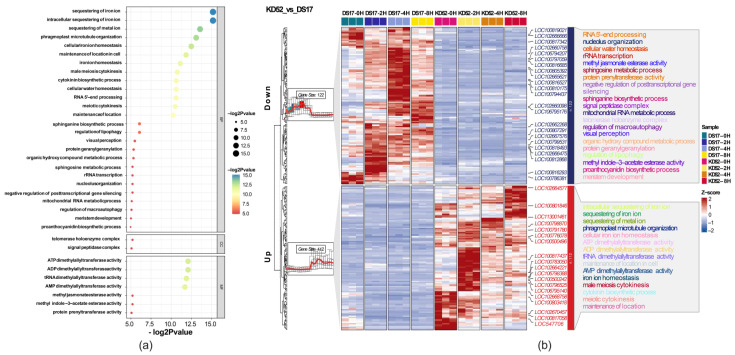
Functional classification and enrichment analysis of differentially expressed genes (DEGs) during cold stress. (**a**) Bubble chart depicting gene ontology (GO) terms enriched in 234 cold-responsive DEGs. The x-axis represents the negative log of *p*-values, and the y-axis displays GO categories, with bubble size indicating the number of DEGs associated with each term. The gradient color scheme indicates the magnitude of the *p*-values, with blue colors representing higher significance and red colors indicating lower significance. (**b**) Clustering heatmap illustrating DEG expression patterns during cold exposure. Of the DEGs, 112 were up-regulated and 122 were down-regulated in the cold-tolerant KD52 compared with the cold-sensitive DS17 at 0, 2, 4, and 8 h. A total of 35 enriched GO terms were identified, comprising 15 up-regulated and 20 down-regulated categories.

**Figure 4 biology-13-00856-f004:**
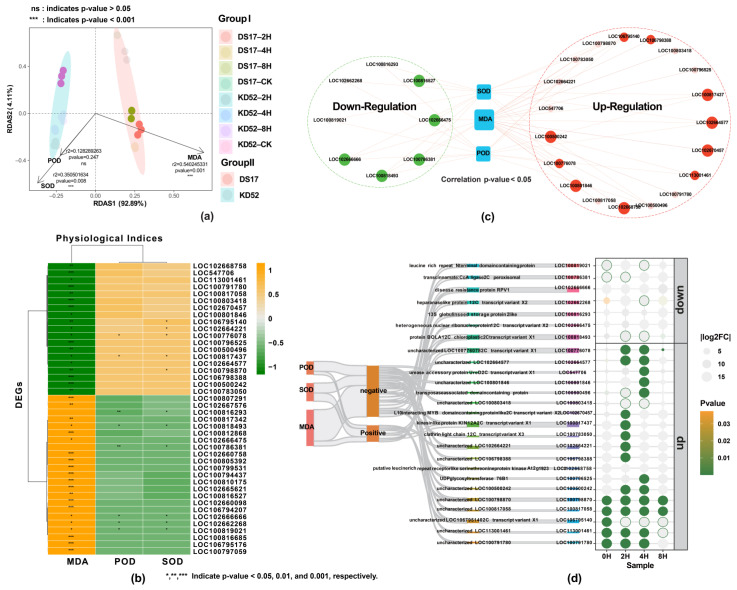
Association analysis between key DEGs and physiological indices. (**a**) Redundancy analysis (RDA) based on the DEGs of samples under the cold stress (Mantel test, n = 24, *p*-value < 0.05). (**b**) Correlations between 42 key DEGs and physiological indices. The pairwise comparisons of a physiological index are shown, with a color gradient denoting Spearman’s correlation coefficient. (**c**) Co-expression network analysis of DEGs and physiological indices during soybean cold stress. Up-regulated DEGs are shown as red circles, and down-regulated DEGs as green circles. Blue rectangles denote the physiological parameters POD, SOD, and MDA. Lines indicate correlations between DEGs and physiological indices, where their thickness indicates the strength of the correlation. The network topology reveals associations between gene expression changes and alterations in physiological parameters. (**d**) Sankey diagram representing the contributions of a physiological index to DEGs. The four columns (from left to right) are physiological index, correlation (Spearman’s *p*-value < 0.05), function, and DEGs’ names at different times (0, 2, 4, and 8 h) in cold stress. *, **, and *** indicate significant difference at *p*-value of <0.05, <0.01, and <0.001, respectively.

**Figure 5 biology-13-00856-f005:**
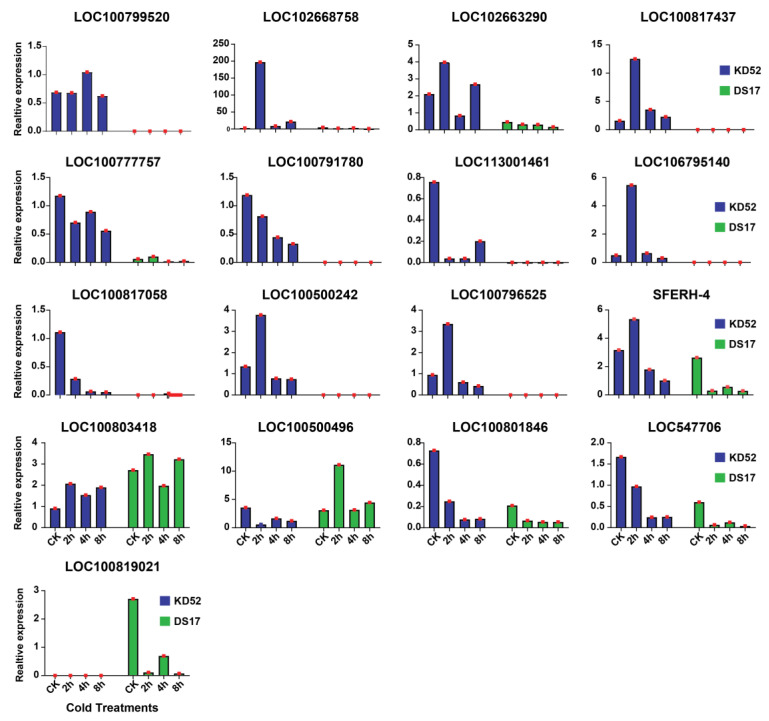
Expression dynamics of cold-responsive genes in soybean cultivars under progressive cold stress. The graphs depict relative transcript levels for 17 genes in KD52 and DS17 cultivars, comparing control conditions (CK, 0 h) with cold treatments lasting 2, 4, and 8 h. Relative transcript abundance represents gene expression levels.

**Figure 6 biology-13-00856-f006:**
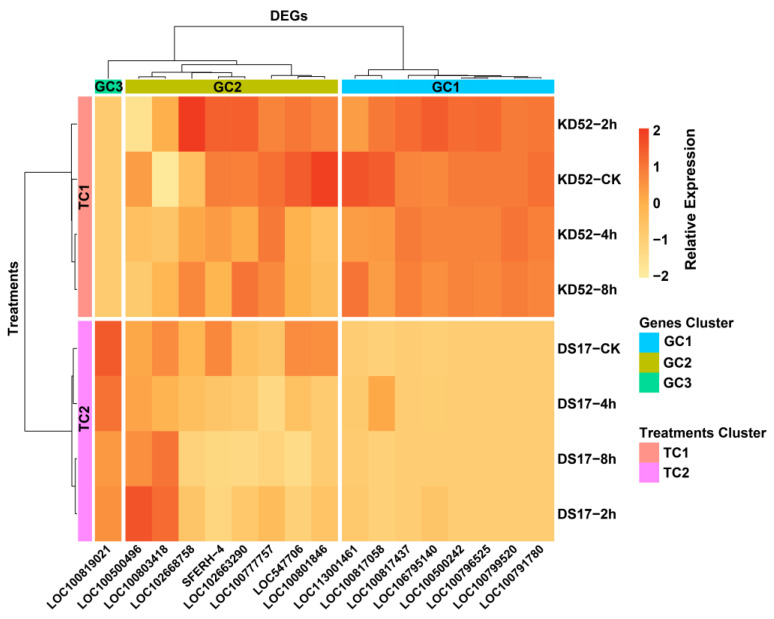
Expression profiling of cold-responsive genes in soybean cultivars subjected to varying stress conditions. The heatmap depicts the relative expression levels (log2-transformed) of 17 genes in cultivars KD52 and DS17 under control (CK, 0 h) and cold treatment (2, 4, and 8 h) conditions. The intensity of the color reflects the magnitude of the change. Hierarchical clustering classified the genes into three co-expressed groups (GC1-GC3) and treatments into two main clusters based on expression patterns (TC1 and TC2).

## Data Availability

The derived data supporting the findings of this study are available from the corresponding authors upon reasonable request.

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
