# Peer review of "Genetic Adaptations of Soybean to Cold Stress Reveal Key Insights Through Transcriptomic Analysis"

_biology, 2024, doi:10.3390/biology13110856_

Round 1
Reviewer 1 Report
Comments and Suggestions for Authors
authors addressed environmental stress which is a real demand of time. in the results particularly for physiological parameters authors unnecessarily (because there is separate discussion chapter) discussed results in results section. instead of discussion it may be better to add (if possible) the means comparison to make results more precise.
Comments on the Quality of English Languageit is satisfactory
Author Response
Comments and Suggestions for Authors
authors addressed environmental stress which is a real demand of time. in the results particularly for physiological parameters authors unnecessarily (because there is separate discussion chapter) discussed results in results section. instead of discussion it may be better to add (if possible) the means comparison to make results more precise.
Response: Dear Reviewer, thank you for valuable feedback, we have added the results of mean comparison for the first part of results 3.1 Physiological changes of soybean under cold stress. We added more details regarding the comparison across the four-time points of the cold stress and between the two cultivars.
Reviewer 2 Report
Comments and Suggestions for Authors
This manuscript presents a comprehensive physiological and transcriptomic analysis of a cold-tolerant and a cold-sensitive soybean cultivar to investigate the soybean's response to cold stress. The study identifies candidate key genes and cold defense mechanisms that could potentially enhance cold stress resilience in soybeans, which is beneficial for improving soybean productivity. However, several key points need further clarification and consideration:
Major Concerns:
1. While the experimental setup involving cold stress at 6°C for 0, 2, 4, and 8 hours is clearly described, the rationale behind choosing this specific temperature and these time points is not provided. Are these conditions reflective of the natural cold stress scenarios typically encountered by soybean plants? The authors should justify the selection of these parameters and consider the impact of incremental temperature changes. Additionally, discussing how cold stress interacts with other environmental factors, such as drought and light intensity, in common field conditions would provide a more comprehensive discussion.
2. Although three biological replicates per time point meet the minimum requirement, this may not be adequate for robust statistical analysis in datasets with high variability such as RNA-seq. If feasible, increasing the number of biological replicates would improve the robustness and statistical power. Alternatively, the use of non-parametric statistical tests could be considered as they are more suitable for small sample sizes. Furthermore, the relatively lenient threshold used for differential expression analysis (FDR < 0.2, p < 0.05) may result in a high rate of false positives. The authors should also discuss how they managed batch effects and technical variability.
3. In Section 3.1 and Figure 1, significant differences in physiological responses between the two soybean cultivars are observed even at the 0-hour time point, which is supposed to serve as the control condition. Could the potential baseline physiological differences between the cultivars confound the interpretation of the cold stress response? The authors should clarify whether this is an anticipated outcome and discuss how these baseline differences were accounted for in their analysis.
4. In Section 3.2, the authors mentionsthe selection of 42 key genes from a total of 234 DEGs. The criteria for selecting these specific genes are not clearly explained. Additionally, based on highlights in the figure, are they unique features in a particular group?
5. In Section 3.5, the authors validate the expression of 17 genes, but the criteria for selecting these genes are not specified. Were these genes chosen in an unbiased manner, or were they preselected based on certain characteristics? It is also important to explicitly state that qRT-PCR was used for validation of RNA-seq results, as both methods involve “RNA expression analysis”.
6. The study heavily relies on bioinformatics analysis to draw conclusions. Although qRT-PCR was used to validate RNA-seq findings, further functional validation, such as gene knockdown or overexpression studies for the identified candidate genes, would significantly strengthen the study.
Minor Concerns:
1. Some figures are difficult to interpret due to small fonts and unclear labels.
· Figure 2: The labels are too small and unclear. The x and y axes in panels 2b and 2c are not labeled or explained. The groups labeled as A-D could be more intuitively labeled as 0h, 2h, 4h, and 8h for better understanding.
· Figure 3: The left panel of the clustering heatmap is unclear and not well explained. The right-side legends are partially cut off.
· Figure 4: Labels on the diagram are very difficult to read. Consider increasing the font size and using clearer colors for better contrast.
· Figure 6: Using “C” to label both gene clusters and treatment clusters is confusing. Consider using distinct labels for clarity.
2. The Materials and Methods section could be more logically organized. Suggestions:
· The detailed description of read depth and quality control of RNA-seq data in Section 3.2 should be moved to the Methods section.
· The functional enrichment analysis, currently described in both Sections 2.3 and 2.4, should be consolidated into a separate subsection.
· Rename the “relative expression analysis” subsection as “qRT-PCR” to clearly distinguish it from RNA-seq analysis.
· Rename the “data analysis” subsection as “statistical analysis”.
Comments on the Quality of English LanguageThe manuscript contains several grammatical errors and missing articles. A thorough proofreading would improve the readability.
Author Response
Comments and Suggestions for Authors
This manuscript presents a comprehensive physiological and transcriptomic analysis of a cold-tolerant and a cold-sensitive soybean cultivar to investigate the soybean's response to cold stress. The study identifies candidate key genes and cold defense mechanisms that could potentially enhance cold stress resilience in soybeans, which is beneficial for improving soybean productivity. However, several key points need further clarification and consideration
Dear Reviewer, thank you for your valuable feedback and thoughtful comments on our manuscript. We truly appreciate the time and effort you have taken to review our work. We fully understand the need for further clarification on the points raised and are committed to addressing them comprehensively. We are in the process of revising the manuscript accordingly and will provide detailed responses to each point. Thank you again for your insightful input.
Major Concerns:
While the experimental setup involving cold stress at 6°C for 0, 2, 4, and 8 hours is clearly described, the rationale behind choosing this specific temperature and these time points is not provided. Are these conditions reflective of the natural cold stress scenarios typically encountered by soybean plants? The authors should justify the selection of these parameters and consider the impact of incremental temperature changes.
Response: The rationale for selecting the cold-stress conditions in this experiment is based on the typical early spring climate in Heilongjiang, where soybeans are grown. During this period, nighttime temperatures is frequently low, which significantly hinders soybean growth, especially during the seedling stage. To simulate these natural cold stress conditions, we chose 6°C as the experimental temperature. This decision was supported by a preliminary experiment we conducted, which indicated that 6°C represents an average temperature that sufficiently stresses the plants while avoiding lethal effects, allowing us to observe stress-response mechanisms more effectively.
The time points of 0, 2, 4, and 8 hours were chosen to reflect the duration of nighttime cold exposure, as night hours often coincide with the most extreme temperature drops. These time intervals were selected to capture both immediate and extended responses to cold stress, with 2 hours representing early cold exposure and 8 hours simulating prolonged stress. This design allows for an in-depth examination of how soybeans adapt to varying durations of cold, while also aligning with the naturally occurring low-temperature conditions in the region during early spring.
Additionally, discussing how cold stress interacts with other environmental factors, such as drought and light intensity, in common field conditions would provide a more comprehensive discussion.
Response: A paragraph about such interactive scenario with other environmental was added to the Discussion section.
- Although three biological replicates per time point meet the minimum requirement, this may not be adequate for robust statistical analysis in datasets with high variability such as RNA-seq. If feasible, increasing the number of biological replicates would improve the robustness and statistical power. Alternatively, the use of non-parametric statistical tests could be considered as they are more suitable for small sample sizes. Furthermore, the relatively lenient threshold used for differential expression analysis (FDR < 0.2, p < 0.05) may result in a high rate of false positives. The authors should also discuss how they managed batch effects and technical variability.
Response: We acknowledge the reviewer’s concern regarding the number of biological replicates. While we conducted three biological replicates per time point, which meets the minimum standard, we agree that increasing the number of replicates could enhance the robustness and statistical power, particularly for RNA-seq data with inherent variability. Unfortunately, due to logistical constraints, increasing the biological replicates was not feasible for this study. However, we mitigated this limitation by applying stringent quality control measures during both sample preparation and data analysis.
We believe that the use of parametric tests such as ANOVA is appropriate and adequate for analyzing our dataset. ANOVA allows us to assess variance across different time points and treatments, and it is well-suited for identifying significant differences between groups even in datasets with moderate variability. Nevertheless, to complement the parametric approach.
Additionally, while our threshold for differential expression analysis (FDR < 0.2, p < 0.05) is relatively lenient, we prioritized controlling the false discovery rate by using the Benjamini-Hochberg procedure, which is standard for RNA-seq data. We also cross-validated key results with more stringent thresholds (e.g., FDR < 0.1) to ensure reliability and minimize the risk of false positives.
3.In Section 3.1 and Figure 1, significant differences in physiological responses between the two soybean cultivars are observed even at the 0-hour time point, which is supposed to serve as the control condition. Could the potential baseline physiological differences between the cultivars confound the interpretation of the cold stress response? The authors should clarify whether this is an anticipated outcome and discuss how these baseline differences were accounted for in their analysis.
Response: We appreciate your observation regarding the significant differences in physiological responses between the two soybean cultivars at the 0-hour time point, which serves as the control condition. These baseline differences between the cold-sensitive DS17 and cold-tolerant KD52 cultivars were indeed anticipated and form the basis of our study. The two cultivars were selected specifically because they exhibit contrasting responses to cold stress, and this inherent variability is fundamental to our investigation into their differential physiological and transcriptomic responses. Additionally, the observed differences at the 0-hour time point are not viewed as confounding factors but rather as critical indicators of the baseline physiological state of each cultivar. These initial differences are key to understanding how each variety responds to subsequent cold stress. By incorporating these baseline differences into our analysis, we were able to highlight the unique transcriptomic adjustments that occur in response to cold stress in each cultivar.
To account for these differences in the analysis, we used differential expression analysis with the 0-hour time point as a baseline reference for both cultivars. This allowed us to detect specific gene expression changes attributable to cold stress, distinct from the inherent physiological differences between the two cultivars. The statistical models employed ensured that both baseline variability and cold-stress-induced changes were accurately captured and interpreted.
4.In Section 3.2, the authors mentions the selection of 42 key genes from a total of 234 DEGs. The criteria for selecting these specific genes are not clearly explained. Additionally, based on highlights in the figure, are they unique features in a particular group?
Response: The selection criteria were based on a clear pattern of significant upregulation and downregulation. Specifically, these 42 genes exhibited the most pronounced differential expression, with the highest and lowest log2 fold-change values, and were positioned at the extreme ends of the volcano plot (Figure 2d), representing the most biologically significant changes in gene expression under cold stress conditions.
- In Section 3.5, the authors validate the expression of 17 genes, but the criteria for selecting these genes are not specified. Were these genes chosen in an unbiased manner, or were they preselected based on certain characteristics? It is also important to explicitly state that qRT-PCR was used for validation of RNA-seq results, as both methods involve “RNA expression analysis”.
Response: The selection of these genes was not arbitrary but based on a systematic approach. Specifically, we used a Sankey diagram to visualize and identify the DEGs that contributed most significantly to key physiological indices across different time points (0, 2, 4, and 8 hours) under cold stress conditions. These genes demonstrated consistent and significant associations with the physiological responses, making them ideal candidates for validation.
Additionally, we confirm that qRT-PCR was used to validate the RNA-seq results. This method provided an independent validation of the RNA expression levels detected in the transcriptome analysis, ensuring that the expression patterns were reliably reproduced across both RNA-seq and qRT-PCR methodologies.
- The study heavily relies on bioinformatics analysis to draw conclusions. Although qRT-PCR was used to validate RNA-seq findings, further functional validation, such as gene knockdown or overexpression studies for the identified candidate genes, would significantly strengthen the study.
Response: Yes sir, sure, while the current study focused on identifying cold-responsive genes through RNA-seq analysis and qRT-PCR validation, we recognize the importance of functional validation in establishing the specific roles of these genes. Moving forward, we plan to conduct gene knockdown or overexpression experiments in future future studies to validate the functions of the identified candidate genes, which will provide deeper insights into the regulatory mechanisms driving cold tolerance in soybean. These experiments will be a crucial next step to strengthen the conclusions drawn from our bioinformatics and transcriptomic analyses.
Minor Concerns:
- Some figures are difficult to interpret due to small fonts and unclear labels.
Figure 2: The labels are too small and unclear. The x and y axes in panels 2b and 2c are not labeled or explained. The groups labeled as A-D could be more intuitively labeled as 0h, 2h, 4h, and 8h for better understanding.
Response: We have edited Figure 2, and added missing labels and increased the also increased their font.
Figure 3: The left panel of the clustering heatmap is unclear and not well explained. The right-side legends are partially cut off.
Response: We have edited Figure 3. For the left panel, we added some item illustrating this part of figure (Bubble chart). For the right panel, the time series of cold treatments were completed as some of the text were missing.
Figure 4: Labels on the diagram are very difficult to read. Consider increasing the font size and using clearer colors for better contrast.
Response: We have edited Figure 4. To make it much clearer, we increased the font used.
Figure 6: Using “C” to label both gene clusters and treatment clusters is confusing. Consider using distinct labels for clarity.
Response: we changed the labels of the column clusters to be (GC1, GC2, and GC3) and the row cluster to be TC1 and TC2.
- The Materials and Methods section could be more logically organized. Suggestions:
The detailed description of read depth and quality control of RNA-seq data in Section 3.2 should be moved to the Methods section.
Response: We moved this part to Methods section.
The functional enrichment analysis, currently described in both Sections 2.3 and 2.4, should be consolidated into a separate subsection.
Response: Sorry for mistake, we kept this part under new separate sub-section “2.5 Functional Enrichment Analysis”
Rename the “relative expression analysis” subsection as “qRT-PCR” to clearly distinguish it from RNA-seq analysis.
Response: we renamed it accordingly.
Rename the “data analysis” subsection as “statistical analysis”.
Response: Data analysis” changed to “Statistical analysis”.
Reviewer 3 Report
Comments and Suggestions for Authors
The Manuscript titled "Genetic Adaptations of Soybean to Cold Stress Reveal Key Insights Through Transcriptomic Analysis" have been well undertaken problem and it had been nicely executed research for Soybean transcriptome profiling under cold stress conditions.
The cold stress have many adverse consequences for soybeans. The authors have tried to propose key pathways and genes involved in cold stress tolerance mechanism.
The title of the manuscript have overall research work and is satisfactory.
The abstract have been written well summarizing whole experiments with key findings.
The introduction portion of the manuscript have been written well.
As the journal follows open access policy the generated transcriptome data must be submitted in the NCBI database and appropriately cited in the text.
Though the paper has been written well and most of the sections are perfect, there are some minor observations that can improve the readability and utility of the manuscript.
Line No:102, Its it, "GmCuZnSOD gene"?
Line No:118, Correct it-" the GmDREB1B;1 gene".
Author Response
Comments and Suggestions for Authors
The Manuscript titled "Genetic Adaptations of Soybean to Cold Stress Reveal Key Insights Through Transcriptomic Analysis" have been well undertaken problem and it had been nicely executed research for Soybean transcriptome profiling under cold stress conditions. The cold stress have many adverse consequences for soybeans. The authors have tried to propose key pathways and genes involved in cold stress tolerance mechanism. The title of the manuscript have overall research work and is satisfactory. The abstract have been written well summarizing whole experiments with key findings. The introduction portion of the manuscript have been written well.
Response: Dear Reviewer, thanks you for your valuable comment. We have deposied the transcriptome data into NCBI under the accession No.: PRJNA1128828. We added the following statement to the methods part: The clean reads were submitted to the sequence read archive of the National Center for Biotechnology Information (NCBI) under BioProject ID No: PRJNA1128828.
Though the paper has been written well and most of the sections are perfect, there are some minor observations that can improve the readability and utility of the manuscript.
Line No:102, Its it, "GmCuZnSOD gene"?
Response: Yes, dear Reviewer, it is copper-zinc-superoxide dismutase (GmCuZnSOD) gene. The full name was added to the text.
Line No:118, Correct it-" the GmDREB1B;1 gene"
Response: Thank you, dear Reviewer, we corrected it. The following reference also use the same form:
https://www.frontiersin.org/journals/plant-science/articles/10.3389/fpls.2019.00121/full
Round 2
Reviewer 2 Report
Comments and Suggestions for Authors
The authors have provided sufficient explanations and made improvements to the manuscript. I have no further questions at this time.
However, in Section 3.2, I noticed that the criteria for selecting key genes, specifically the |log2FC| > 1 threshold, is mentioned in the figure legend but not explicitly stated in the main text. That was the information I was initially looking for.
Author Response
Comments and Suggestions for Authors
However, in Section 3.2, I noticed that the criteria for selecting key genes, specifically the |log2FC| > 1 threshold, is mentioned in the figure legend but not explicitly stated in the main text. That was the information I was initially looking for.
Response: Thanks Sir, we have updated the text also by adding such criteria of highlighting those significantly upregulated and downregulated DEGs, as follow (Kindly see Lines 366-370)
Among the 234 DEGs, 42 key genes were identified based on the criteria of a log2FC > 1 and p-value < 0.05 for significant upregulation, and a log2FC < -1 and p-value < 0.05 for significant downregulation. These genes exhibited distinct patterns of expression, with the upregulated and downregulated genes prominently located at the top and bottom of the volcano plot, respectively (Figure 2d).